# Anti-SARS-CoV-2 Omicron Antibodies Isolated from a SARS-CoV-2 Delta Semi-Immune Phage Display Library

**DOI:** 10.3390/antib11010013

**Published:** 2022-02-10

**Authors:** Ivette Mendoza-Salazar, Keyla M. Gómez-Castellano, Edith González-González, Ramsés Gamboa-Suasnavart, Stefany D. Rodríguez-Luna, Giovanni Santiago-Casas, María I. Cortés-Paniagua, Sonia M. Pérez-Tapia, Juan C. Almagro

**Affiliations:** 1Unidad de Desarrollo e Investigación en Bioterapéuticos (UDIBI), Escuela Nacional de Ciencias Biológicas, Instituto Politécnico Nacional, Mexico City 11340, Mexico; ivette.mendoza@udibi.com.mx (I.M.-S.); keyla.gomez@udibi.com.mx (K.M.G.-C.); edith.gonzalez@udibi.com.mx (E.G.-G.); ramses.gamboa@udibi.com.mx (R.G.-S.); stefany.luna@udibi.com.mx (S.D.R.-L.); giovanni.santiago@udibi.com.mx (G.S.-C.); ilselena.cortes@udibi.com.mx (M.I.C.-P.); 2Departamento de Inmunología, Escuela Nacional de Ciencias Biológicas, Instituto Politécnico Nacional (ENCB-IPN), Mexico City 11340, Mexico; 3Laboratorio Nacional para Servicios Especializados de Investigación, Desarrollo e Innovación (I+D+i) para Farmoquímicos y Biotecnológicos, LANSEIDI-FarBiotec-CONACyT, Mexico City 11340, Mexico; 4GlobalBio, Inc., 320 Concord Ave, Cambridge, MA 02138, USA

**Keywords:** COVID-19, receptor-binding domain (RBD), therapeutic antibodies, phage display, VOCs

## Abstract

This report describes the discovery and characterization of antibodies with potential broad SARS-CoV-2 neutralization profiles. The antibodies were obtained from a phage display library built with the VH repertoire of a convalescent COVID-19 patient who was infected with SARS-CoV-2 B.1.617.2 (Delta). The patient received a single dose of Ad5-nCoV vaccine (Convidecia™, CanSino Biologics Inc.) one month before developing COVID-19 symptoms. Four synthetic VL libraries were used as counterparts of the immune VH repertoire. After three rounds of panning with SARS-CoV-2 receptor-binding domain wildtype (RBD-WT) 34 unique scFvs, were identified, with 27 cross-reactive for the RBD-WT and RBD Delta (RBD-DT), and seven specifics for the RBD-WT. The cross-reactive scFvs were more diverse than the RBD-WT specific ones, being encoded by several IGHV genes from the IGHV1 and IGHV3 families combined with short HCDR3s. Six cross-reactive scFvs and one RBD-WT specific scFv were converted to human IgG1 (hIgG1). Out of the seven antibodies, six blocked the RBD-WT binding to angiotensin converting enzyme 2 (ACE2), suggesting these antibodies may neutralize the SARS-CoV-2 infection. Importantly, one of the antibodies also recognized the RBD from the B.1.1.529 (Omicron) isolate, implying that the VH repertoire of the convalescent patient would protect against SARS-CoV-2 Wildtype, Delta, and Omicron. From a practical viewpoint, the triple cross-reactive antibody provides the substrate for developing therapeutic antibodies with a broad SARS-CoV-2 neutralization profile.

## 1. Introduction

COVID-19 is caused by the severe acute respiratory syndrome coronavirus 2 (SARS-CoV-2). This coronavirus is made of sixteen non-structural proteins and four structural proteins. The non-structural proteins are involved mainly in virus replication [1,2]. The structural proteins, including the membrane (M), nucleocapsid (N), envelope (E) and spike (S) proteins, are implied in the assembly and infection of the virus [3,4]. Protein S has two subunits, S1 and S2, which are responsible for the specific recognition of human angiotensin converting enzyme 2 (hACE2) on the surface of host cells and viral entry. S1 specifically binds the hACE2 through its receptor-binding domain (RBD), whereas S2 facilitates the fusion of the virus to the cell membrane [5]. Most of the natural neutralizing antibody responses to SARS-CoV-2 are mounted against the S1 protein, and more specifically, against the RBD, making these proteins the main targets for vaccine and therapeutic antibody development [6].

Twelve COVID-19 vaccines have been granted with approval or emergency use authorization (EUA) in a variety of countries [7], and over two hundred vaccines are in preclinical development and clinical trials [8,9]. Nonetheless, the global demand for vaccines has exceeded the number that can be manufactured, thus leading to vaccine shortages, particularly in developing countries. Moreover, SARS-CoV-2 variants, called variants of concern (VOCs), in particular the B.1.617.2 (Delta) isolate [10], and more recently, B.1.1.529 (Omicron) [11,12], have compromised the effectiveness of some vaccines [13,14,15]. Furthermore, vaccines only offer a prophylactic solution to control the pandemic, whereas individuals already infected or those who do not properly respond to vaccination need therapeutic alternatives to recover from the infection.

Antibodies play the dual role of prophylactic and therapeutic treatments, with the advantage over vaccines of faster development and approval processes. In fact, the first cocktail of therapeutic antibodies to treat COVID-19, named REGEN-COV (casirivimab with imdevimab), received US Food and Drug Administration (FDA) EUA for medical use in humans in November 2020, less than a year after the World Health Organization (WHO) declared COVID-19 a pandemic and before any COVID-19 vaccine received FDA or EMA EUA [16]. Four additional antibody-based drugs have received EUA by the FDA and/or EMA, including: (1) a cocktail of bamlanivimab and etesevimab developed by Eli Lilly [17]; (2) sotrovimab, developed and marketed by Glaxo Smith Klein; (3) regdanvimab, commercialized by Celltrion; and more recently (4) AstraZeneca’s Evusheld, which contains tixagevimab co-packaged with cilgavimab [18]. Dozens of other neutralizing antibodies are in preclinical and clinical trials, targeting diverse SARS-CoV-2 proteins [19]. Yet, some of the FDA- and EMA-approved antibodies and some in development have limited applications since they have failed to recognize VOCs [15,20]. This, compounded with the fact that SARS-CoV-2 will continue to evolve and very likely generate other immune escape variants, as shown by the emergence of Omicron [21], has led to a continuous race for developing new and more effective therapeutic antibodies in unprecedented time.

Here, with the two-fold aim of exploring the immune response to SARS-CoV-2 Delta and discovering antibodies recognizing diverse SARS-CoV-2 VOCs, we built a semi-immune scFv phage display library with the VH repertoire of a convalescent COVID-19 patient infected with SARS-CoV-2 Delta, who was previously vaccinated with a single dose of Convidecia™. As counterpart of the repertoire of VH chains, four synthetic VL libraries described elsewhere (manuscript in preparation) were used. After three rounds of panning using RBD wildtype (RBD-WT) as a selector, antibodies blocking the RBD-WT:hACE2 interaction were obtained. One of them cross-reacted with RBD WT, Delta (RBD-DT) and Omicron (RBD-O). The implications of these findings for gaining insight into the antibody immune response to SARS-CoV-2 VOCs, as well as developing therapeutic antibodies with a broad SARS-CoV-2 neutralization profile, are discussed.

## 2. Materials and Methods

### 2.1. Immune VH Repertoire

Peripheral blood mononuclear cells (PBMCs) were obtained from a convalescent COVID-19 patient with a high titer of anti-RBD IgG antibodies. The patient experienced mild symptoms of COVID-19 infection and was diagnosed with SARS-CoV-2 by RT-PCR. The viral strain of SARS-CoV-2 was determined by complete genome sequencing (Genbank accession number: OM060237) of RNA isolated from a nasopharyngeal swab taken in the acute phase of COVID-19 infection. The patient reported having received a single dose of Convidecia™ one month prior to developing COVID-19 symptoms (July 2021). The blood sample was collected five weeks post-infection under written informed consent.

Starting from the PBMCs, total RNA was isolated using Trizol (Invitrogen, Burlington, Canada, Cat. Nos.: 15596026 and 15596018) and used as substrate to generate messenger RNA (mRNA) via the polyA Spin™ mRNA Isolation Kit (New England Biolabs, Ipswich, MA, USA, Cat No.: S1560S) following the manufacturer’s instructions. The cDNA was generated via reverse transcription with the Protoscript^®^ II First Strand cDNA synthesis Kit (New England Biolabs, Ipswich, MA, USA, Cat. No.: E6560S) and a poly-T oligo. Double-stranded DNA encoding the VH repertoire was PCR-amplified with the primers described by Noy-Porat et al., 2020 [22], with some minor modifications to adapt the primers to cloning the PCR fragments containing the VH repertoire into our phage display vector. The quality of the VH fragments was assessed by cloning an aliquot of the PCR product into a TOPO vector (ThermoFisher Scientific, Carlsbad, CA, USA, Cat No.: K4575J10) and sequencing 10 clones. The sequences indicated that all the VH fragments were different, representing several VH gene families and with a length variation at the HCDR3 and sequences that resembled the human HCDR3 repertoire.

### 2.2. Assemblage with Four Synthetic VL Fragments

The immune VH repertoire was combined with the four synthetic VL libraries of ALTHEA Gold Plus Libraries™ (manuscript in preparation). These VL libraries were built with the 3-20/4 and 4-01/4 scaffolds from ALTHEA Gold Libraries™ [23] and two additional scaffolds: 3–11/4 and 1-39/4. The 4-01/4 scaffold has a long LCDR1 loop and the other three (3-20/4, 3-11/4 and 1-39/4) have short LCDR1 loops. Within the scaffolds with short LCDR1 loops, 3-20/4 has the canonical structure class 6-1-1, whereas 3-11/4 and 1-39/4 have the canonical structure class 2-1-1 [24], thus providing structural diversity to the set of VL scaffolds. The human genes that severed as a template to design these VL scaffolds are the most prevalent in immune responses to diverse targets and have frequently been used as scaffolds to build antibody phage displayed libraries [24,25,26].

### 2.3. Phage Display scFv Library Construction

The repertoire of VH fragments was independently combined with each of the four synthetic VLs (four libraries total) in a VL-linker-VH configuration via a GS19 linker peptide by overlapping PCR. The resultant scFvs were digested with SfiI (New England Biolabs, Ipswich, MA, USA, Cat. No.: R0123L) and ligated with the pADL™ 23c phagemid vector (Antibody Design Lab, San Diego, CA, USA). Five micrograms of each ligated product were electroporated into electrocompetent TG1 cells. The size of the libraries was estimated in 10^8^–10^9^ colony-forming units (cfus). The quality of the libraries was assessed by sequencing five clones from each library (20 clones in total). All the clones had unique VH sequences coming from diverse human IGHV gene families combined with unique HCDR3 sequences. All the sequenced clones had variants of the synthetic VL scaffolds and the proper scFv configuration.

### 2.4. Expression and Purification of SARS-CoV-2 RBD Recombinant Proteins

The expression, purification, and characterization of the RBD-WT and RBD-DT have been described elsewhere [27]. In brief, the plasmid containing the RBD-WT (or mutated in our laboratory to obtain the RBD-DT) donated by Dr Florian Kammer at Department of Microbiology, Icahn School of Medicine at Mount Sinai, New York, NY, USA. was expanded in the *E. coli* DH5α strain. Purified plasmid DNA was used to transfect HEK 293T cells (ATCC CRL-3216). The transfected cells were incubated four days at 37 °C in 5% CO_2_. The RBD was purified by immobilized metal affinity chromatography (IMAC) using 5 mL-HIS Trap™ Nickel columns (GE Healthcare, Darmstadt, Germany, Cat. No.: 17-5255-01). The quality of the purified RBD was assessed by SEC-HPLC, SDS-PAGE and ELISA with commercially available anti-RBD antibodies. The RBD Omicron was purchased at SinoBiological (Wayne, NJ, USA, Cat. No.: 40592-V08H121).

### 2.5. Phage-Antibody Selection

To subtract unspecific phages, 100 µL of streptavidin-magnetic beads (Dynabeads™ M-280 Streptavidin, Thermo Fisher Scientific, Rochester, NY, USA) were incubated with each of the libraries. Biotinylated RBD-WT in PBS-BSA 3%, at decreasing concentrations of 50, 10 and 2 nM, was incubated at room temperature (RT) for one hour with the libraries for rounds 1, 2 and 3, respectively. The first round of selection was performed with 5 × 10^12^ virions of each the four libraries independently. The subsequent rounds were performed by mixing the output of the previous round and incubating 1 × 10^12^ virions with the libraries after rescuing them with helper phage CM13K (ADL; Cat. No.: PH050L). The RBD-phage complexes were precipitated with pre-treated streptavidin-magnetic beads. The unbound phages were washed away with PBS-T (PBS—Tween-20 0.1%) and PBS. After the washing steps, specific phages were eluted with a Trypsin-TPCK (Sigma, Toluca, Mexico, Cat. No.: 4370285) at 1 mg/mL for 10 min. A second elution was performed with Glycine-HCl pH 2.2 buffer for additional 10 min.

### 2.6. Expression and Specific Binding to RBD-WT and RBD-DT

Colonies were picked out from 2xYT plates of round 3, incubated in 2 mL Nunc™ DeepWell plates (Thermo Scientific™, Rochester, NY, USA, Cat. No.: 278743) containing 2xYT of glucose (1%) and carbenicillin (100 µg/mL), and grown overnight at 37 °C. ScFv expression was induced with IPTG 1 mM with overnight incubation at 30 °C. The supernatants containing the IPTG-induced scFvs were tested in two primary assays: (1) scFv expression and (2) binding to RBD-WT and RBD-DT.

The scFv expression was assessed by ELISA using a Protein L assay. NUNC MaxiSorp plates were coated with Protein L (1 µg/mL) in coating buffer (BioRad, Berkeley, CA, USA, Cat. No.: BUF030C) overnight at 4 °C. The plates were washed with PBS and blocked with 3% skim milk in PBS-Tween 0.1% (MPBST). Each induced supernatant was added to the Protein L coated wells in a dilution 1:2 in PBS/milk and incubated at RT for one hour. The plates were washed with PBS and bound scFvs were detected with an anti-myc-HRP antibody (Abcam, Cambridge, MA, USA, Cat. No.: ab19312). The assay was revealed with TMB substrate reagent (BD OptEIA, BD Biosciences, San Diego, Ca, USA, Cat. No.: 555214). The color rection was stopped with phosphoric acid 1 M (Abcam, Cambridge, MA, USA, Cat. No.: ab171529) and plates were read at 450 nm with a correction at 570 nm using a EspectraMax M3 microplate reader (Molecular Devices, LLC; San José, CA, USA).

The binding to RBD-WT and RBD-DT was evaluated by ELISA using a 2-fold dilution of IPTG-induced scFv supernatants. To assess specificity, an ELISA with bovine serum albumin (BSA) was performed side-by-side with the RBD ELISAs. Similar to the protein L expression assay, NUNC MaxiSorp plates were coated with RBD-WT or RBD-DT (1 µg/mL) or BSA (100 µg/mL) in PBS overnight at 4 °C. The coated plates were blocked with MPBST at RT for one hour. The assays were revealed and read as the protein L expression assay. All of the clones with a signal of at least 2× the background, were considered positive and submitted to Sanger sequencing to determine unique clones.

### 2.7. Competition with P5E1-A6 and hACE2 for Binding RBD

To determine the competence with a known neutralizing antibody, a sandwich ELISA with P5E1-A6 (see below) was performed as follows. NUNC MaxiSorp plates were coated with P5E1-A6 antibody (1 µg/mL) in PBS overnight at 4 °C. The plates were washed with PBS, and RBD-WT (1 µg/mL) in MPBST was added and incubated at RT for one hour. The plates were washed with PBS and blocked with MPBST. The supernatants containing the IPTG-induced svFvs were dispensed in a 1:2 dilution in PBS/milk and incubated at RT for one hour. The assay was revealed and read as the protein L- and RBD-binding assays.

The competition with hACE2 for binding RBD was implemented in Intellicyt^®^ iQue3 system (Sartorius; Göttingen, Germany). Biotinylated RBD-WT (SPD-C82E9, Acro biosystems Newark, DE, USA) (40 ng/mL) was incubated with SAv (streptavidin) beads (iQue Qbeads^®^ DevScreen, Sartorius) at RT for 40 min. After 2 washing steps with PBS-BSA 1%, vortexing and centrifugation at 12,000 rpm, the supernatants were carefully removed, and the beads were diluted according to the manufacturer’s instructions. Serial dilutions of the IPTG-induced scFv supernatants were mixed 1:1 with 20 µL (50 ng/mL) of biotinylated hACE2 (Acro biosystems, Newark, DE, USA, Cat. No.: AC2-H82E6) and transferred to a 96 V-well plate. RBD-Qbeads (10 µL) were added, and the Qbeads-RBD-hACE2-biotin complex was created with 10 µL of Streptavidin-PE in a 1:500 dilution. The medium fluorescence intensity (MFI) was recorded in the flow cytometer iQue-3. The data were fit to a four-parameter dose–response in GraphPad Prism 9.3.1.

### 2.8. hIgG1 Conversion

The VH and VL regions of the scFvs were amplified by PCR and cloned in TGEX expression vector (Antibody Design Labs, San Diego, CA, USA). HEK 293T cells (ATCC, Manassas, VA, USA, CRL-3216) were co-transfected with plasmid DNA containing heavy and light chains. The transfected cells were incubated for four days in 5% CO_2_ at 37 °C. The antibodies were purified by affinity chromatography using a Protein A column (MabSelect™ Sure™; Cytiva, Sigma Aldrich, Darmstadt, Germany, Cat. No.: 17-5438-01). The column was sanitized with NaOH 0.15 mM at a flow of 0.2 mL/minutes for five minutes and equilibrated with a 10-column volume (CV) of binding buffer (20 mM Na_2_PO_4_, 150 mM NaCl, at pH 7.2) until reaching base line. Capture of the antibodies was performed at a flow of 2.5 mL/minutes followed by washing the column with 20 CV of binding buffer. The elution was performed with 0.1 M acetic acid at pH 2.8. Ultrafiltration columns of 10 kDa (Millipore, Darmstadt, Germany, Cat. No.: PLGC02510) were used to concentrate the eluted protein and change the buffer. The expression and binding assays (above) were performed as for the scFvs but revealed with a goat anti-Human IgG Fc PE (Invitrogen, Waltham, MA, USA, Cat. No.: 12-4998-82).

### 2.9. Developability Assessment

The purified IgGs were subjected to a robust analytical platform to determine their potential for pharmaceutical developability. This platform considers the determination of purity, identity/integrity, and thermal stability by analytical size exclusion chromatography (SEC), denaturing polyacrylamide electrophoresis (SDS-PAGE), and Protein Thermal Shift™ assay. All of these analytical techniques were performed using standard and well-known physicochemical methods for proteins [27,28,29].

### 2.10. Surface Plasmon Resonance (SPR)

The SPR assay was performed in a BIAcore T200 instrument (GE Healthcare, Piscataway, NJ, USA). An anti-human IgG Fc antibody (R&D Systems, Minneapolis, USA, Cat. No.: MAB110) was immobilized on the surface of a CM5 sensor chip (GE Healthcare Piscataway, NJ, USA, Cat No.: BR100530) at 125 RU. Anti-SARS-CoV-2 antibodies were flown for 60 s at a rate of 10 µL/min. After capturing the antibodies, RBD-WT, RBD-DT or RBD-O, in concentrations ranging from 7.8 to 125 nM, were flown over the chip for 120 s at a rate of 10 μL/min followed by 5 min of dissociation time. The running buffer was HBS-Ep+ (Cytiva, Piscataway, NJ, USA, Cat. No.: BR100669), which contains 10 mM HEPES, 150 mM NaCl, 3 mM EDTA, and 0.05% surfactant P20 at pH 7.4. Regeneration to the baseline was achieved by injections of Glycine-HCl 10 mM (pH 1.5). Association (kon) and dissociation (koff) rate constants were calculated by fitting the raw data to a 1:1 model (Langmuir) using BIAevaluation 3.0 software.

### 2.11. Control Antibodies

Two SARS-CoV-2 neutralizing antibodies, CB6 [30] and P5E1-A6, were used as positive controls to set up the assays and as comparators in the characterization of the antibodies obtained in this work. CB6 is the precursor of etesevimab [31], both sharing the same V regions and differing in the Fc regions. P5E1-A6 was isolated and characterized in our laboratory (manuscript in preparation). It recognizes the RBD-WT and RBD-DT and competes with hACE2 for RBD-WT binding. We used the anti-lysozyme antibody D1.3 as a negative control [32]. The variable regions of P5E1-A6, CB6 and D1.3 were cloned as hIgG1, expressed, and purified following the same procedures as the other antibodies described in this work. An anti-VEGFR3 scFv isolated in our laboratory was used as negative control in the scFv screening.

## 3. Results

### 3.1. Selection of Unique RBD-Positive Clones

After three rounds of solution panning using RBD-WT as a selector, 90 clones were evaluated for binding to RBD-WT and RBD-DT. A total of 53 clones (58%) were positive for RBD-WT or RBD-WT and RBD-DT, with 34 scFvs (38%) being unique scFv sequences. The binding of the unique scFvs to RBD-WT and RBD-DT was confirmed side-by-side with a competition sandwich ELISA assay using P5E1-A6 as a capture reagent and a blocking assay of the RBD:hACE2 interaction (Figure 1). Four clusters (C1–C4) of different functional profiles were found in the set of unique scFvs. C1 (scFvs G12–C10) included scFvs specific for RBD-WT, competed with P5E1-A6 and blocked the RBD:hACE2 interaction. C2 (scFvs A2–B8) were RBD-WT/DT cross-reactive clones, competed with P5E1-A6 and blocked the RBD:hACE2 interaction. C3 (scFvs B4–B11) were RBD-WT/DT cross-reactive clones that did not compete with P5E1-A6 but blocked the RBD:hACE2 interaction. C4 (scFvs H2–G3) were RBD-WT/DT cross-reactive clones which did not compete with P5E1-A6 nor blocked the RBD:hACE2 interaction.

### 3.2. Sequence Patterns and Binding Profile of the Selected scFvs

Table 1 summarizes the main sequence features of the scFvs shown in Figure 1. C1′s scFvs have diverse synthetic VL scaffolds paired with a conserved VH chain. This VH sequence is encoded by the IGHV1-69*10 gene with five somatic mutations in HCDR1 and HCDR2 (data not shown) combined with the same long HCDR3; the longest one (22 residues). Since the conserved VH sequence is combined with diverse synthetic VL scaffolds, this is indicative that VH defined the functional profile of the antibodies whereas VL played a secondary and minor role. C2′s scFvs are more diverse, encoded by genes from two IGHV families, IGHV3-53 and IGHV1-24. The IGHV3-53 gene is combined with two relatively short HCDR3 (11 and 12 residues). The IGHV1-24 is combined with the same HCDR3 of 13 residues. C3 and C4 are mostly populated by scFvs encoded by the IGHV1-46 gene combined with 15-residue HCDR3 sequences (eight out of 11 scFvs in C3 and all the sequences in C4 contain the IGHV1-46 gene). All of the scFvs with the IGHV1-46 gene are combined with two HCDR3 sequences of the same length (15 residues). Similar to C1, C2–C4 showed diverse synthetic VL scaffolds, implying that VH plays the main role in defining the functional clusters.

### 3.3. Conversion to hIgG1 and RBD Omicron Binding

To assess the performance of the scFvs in therapeutic format, seven scFvs (highlighted gray in Table 1) were converted to hIgG1. These seven scFvs were chosen mainly from C2, with additional antibodies representing the other three functional clusters. We favored scFvs with diverse IGHV genes, HCDR3 lengths and synthetic VL scaffolds. The expression yield, binding to RBD-WT and RBD-DT, as well as RBD:hACE2 blocking activity were first assessed in the HEK 293T supernatant. The results (Figure 2) confirmed the functional profile seen in the scFvs. All the antibodies cross-reacted with the RBD-WT and -DT except IgG-G12, which, as expected, was RBD-WT-specific. Additionally, consistent with the results obtained at the scFv level, all the antibodies blocked the RBD:hACE2 interaction, except IgG-H2, which did not block the RBD:hACE2 interaction as scFv either.

Based on these results, four IgGs (IgG-A2, IgG-A7, IgG-B1 and IgG-G12) were protein-A-purified and further characterized (Table 2). All of the IgGs showed a developability profile consistent with therapeutic antibodies, i.e., 100% monomeric content after a single-step protein A purification, SDS-PAGE with the expected bands in reducing and non-reducing conditions, and a high thermal stability. All of the purified antibodies blocked the RBD:hACE2 interaction (Figure 3) with IC_50_ values similar to CB6 [30], which was the precursor of etesevimab, one of the therapeutic antibodies with granted FDA EUA [31].

Binding to RBD-WT, RBD-DT, as well as RBD-O, as assessed by ELISA, is shown in Figure 4. All the antibodies except IgG-G12 recognized both RBD-WT and RBD-DT. Importantly, IgG-A7 recognized all the three RBDs, in contrast with the other antibodies, including CB6 and P5E1-A6, which only bound the RBD-WT or cross-reacted with RBD-WT and RBD-DT, but did not recognize RBD-O. Moreover, the affinity of IgG-A7 for the SARS-CoV-2 RBDs as determined by SPR (Biacore) resulted in K_D_ values of 0.68, 0.24 and 8.18 nM for the RBD-WT, -DT and -O, respectively. Since IgG-A7 also blocked the RBD:hACE2 interaction, it is likely that this triple cross-reactive antibody would be a potent neutralizing antibody of all the three SARS-CoV-2 variants.

It should be noted that A7 was found three times in the 90 (3%) screened clones, meaning that it was not a rare event in the VH repertoire of the COVID-19 patient. On the other hand, it had the same VH chain than B8 (Table 1) and shared the same synthetic VL scaffold, 3–20. The difference between A7 and B8 resided in mutations at VL (data not shown). Since VL seems to have a secondary role in determining the functional profile of the antibodies isolated in this work, it could be possible that B8 is also a broadly neutralizing antibody, thus providing further evidence supporting the suggestion that anti-SARS-CoV-2 Omicron-neutralizing antibodies were relatively frequent in the immune repertoire of the convalescent COVID-19 patient.

## 4. Discussion

In the previous sections, we reported the discovery of a panel of 34 anti-SARS-CoV-2 scFvs with diverse sequences and functional profiles. The functionality of these scFvs was classified in four clusters (C1-C4), which represented at least four distinct epitopes. C1 included seven RBD-WT-specific scFvs that shared a conserved VH chain paired with diverse synthetic VLs. The remaining 27 scFvs cross-reacted with RBD-WT and RBD-DT and were classified in three clusters (C2-C4). The scFvs in C2 blocked the RBD:hACE2 interaction and competed with P5E1-A6, whereas the scFvs in C3 did not compete with P5E1-A6 but blocked the RBD:hACE2 interaction. The scFvs in C4 did not compete with P5E1-A6 nor blocked the RBD:hACE2 interaction. Importantly, one of the antibodies belonging to C2, called IgG-A7, also recognized the RBD-O with an affinity in the single nanomolar digit. Since IgG-A7 also competed with hACE2 and P5E1-A6, it may neutralize the SARS-CoV-2 Wildtype, Delta and Omicron variants.

Our findings have several implications for understanding the immune response to VOCs and the development of therapeutic antibodies with a broad SARS-CoV-2 neutralization profile. From an immunological perspective, it has been reported [7] that Convidecia™ has an effectiveness against SARS-CoV-2 Wuhan of 63.7% after 14 days of vaccination and 57.5% at day 28. To the best of our knowledge, the effectiveness of Convidecia™ against SARS-CoV-2 Delta has not been reported. The patient who donated the PBMCs was infected with SARS-CoV-2 Delta a month after receiving Convidecia™. While the vaccine might have attenuated the effect of the infection, it certainly did not prevent it.

The specific scFvs for the RBD from SARS-CoV-2 Wuhan isolated from the patient competed with hACE2 and P5E1-A6 for binding RBD, suggesting that they might be neutralize the SARS-CoV-2 Wuhan strain. All of these scFvs shared the same VH, encoded by the IGHV1-69*10 germline gene with five mutations in the HCDR1 and HCDR2, and the longest HCDR3 (22 residues) was found in the panel of selected scFvs. Long HCDR3 regions have been found in antibodies elicited in response to viral infections and are a hallmark of neutralization [33]. The IGVH1-69 germline gene, on the other hand, has been found in anti-SARS-CoV-2 neutralizing antibodies [34] as well as in other infectious diseases [35]. It has been proposed that IGVH1-69 is a first-responder component of the antibody repertoire that evolved to initiate a rapid defense against infectious agents [36]. Therefore, the specific RBD-WT antibodies described in this work might have been those “first responders” elicited after immunization with the Convidecia™ vaccine.

SARS-CoV-2 Delta infection followed and might have acted as a natural booster, expanding and maturing populations of cross-reactive RBD-WT/DT clones, as well as diversifying the immune response to confront the viral challenge. In fact, the cross-reactive antibodies were encoded by diverse germline genes, e.g., IGHV1-24, IGHV1-46, IGHV3-53 and IGHV3-9. The human gene IGHV1-24 was reported to be specific for the anti-SARS-CoV-2 antibody response [34]. Of note, the VH sequence of A7, which binds all the RBDs has an identity of 94.8% at the amino acid level with respect to IGHV1-24 germline gene configuration. The identification of this triple cross-reactive antibody, which also happened to compete with hACE2 and P5E1-A6 for RBD binding, implies that infection with SARS-CoV-2 Delta might also provide protection against SARS-CoV-2 Omicron infection.

From a practical perspective, IgG-A7 had all the attributes of a good candidate to be further developed in a therapeutic antibody. It had a sub-nanomolar affinity for RBD-WT and RBD-DT, and a single-digit nanomolar affinity for RBD-O. In addition, IgG-A7 showed a 100% monomeric content after Protein A purification and the expected bands in the SDS-PAGE. Moreover, IgG-A7 had two Tms (Table 2), with the second one of 82 °C very likely corresponding to the Fab. This high thermal stability may translate into a high expression yield—notice that IgG-A7 has the highest expression yield of all the antibodies—as well as a high solubility and long-term stability [37]. Considering the fast evolution of SARS-CoV-2 into VOCs, the limited effectiveness of some vaccines to deal with them, and the lack of efficacy of some FDA and EMA EUA antibodies to treat SARS-CoV2 Delta and Omicron, if IgG-A7 demonstrates the ability to neutralize the three SARS-CoV2 strains in vivo, it could certainly be a short-term solution to control the impact of new waves of SARS-CoV-2 infections.

## 5. Patents

A PCT (Patent Cooperation Treaty) protecting the antibody sequences described in the work is in the process of being filed.

## Figures and Tables

**Figure 1 antibodies-11-00013-f001:**
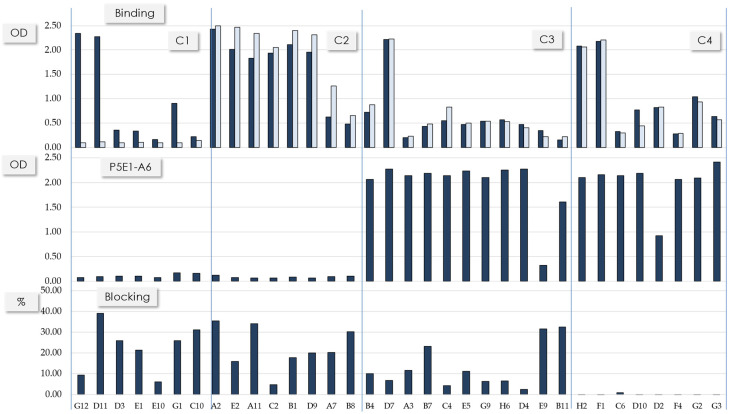
Functional profile of the unique scFvs. Binding to RBD-WT and RBD-DT (top), competition with P5E1-A6 (middle) and RBD-WT:hACE2 blocking interaction (bottom). A scFv was considered positive in the binding assay if it has at lead 2× the negative control OD value. Vertical lines divide the clones into four functional clusters (C1–4).

**Figure 2 antibodies-11-00013-f002:**
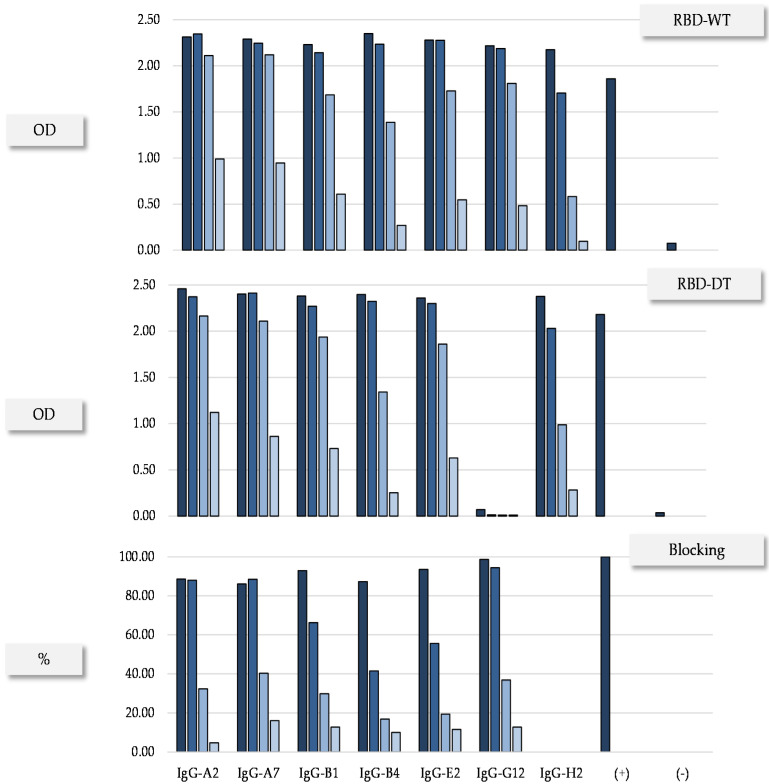
Confirmation of functional profile in HEK 293T of representative IgGs. Binding to RBD-WT, RBD-DT and blocking of the RBD:hACE2 interaction. The supernatants were evaluated in serial dilutions (1:10, 1.100, 1:1000 and 1:10,000) represented in different tones of blue. (+) and (–) indicate positive (P5E1-A6) and negative (D1.3) controls, respectively.

**Figure 3 antibodies-11-00013-f003:**
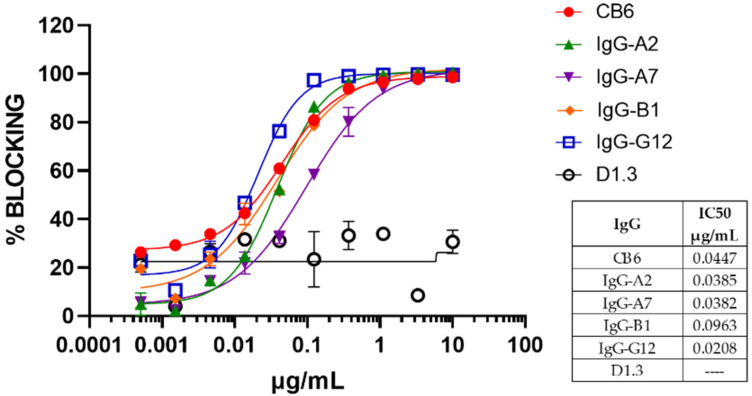
RBD:hACE2 blockade assay for representative IgGs. All IgGs are blocking the RBD:hACE2 interaction with a dose–response behavior in a concentration interval of 0.001–10 μg/mL. The data were fitted to a four-parameter dose–response curve using GraphPad Prism 9.3.1.

**Figure 4 antibodies-11-00013-f004:**
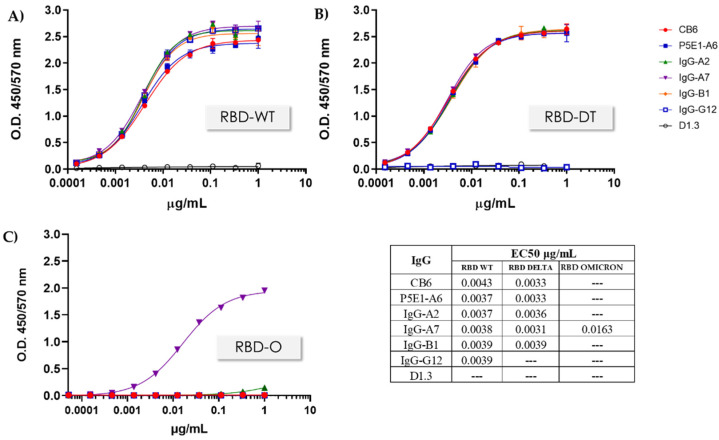
RBD:IgG binding assay. Binding activity to RBD-WT, RBD-DT and RBD-O was assessed for representative IgGs in a concentration interval of 0.0001–1 μg/mL. (**A**) Binding to RBD-WT, (**B**) Binding to RBD-DT and (**C**) Binding to RBD-O. The data were fit to a four-parameter dose–response in GraphPad Prism 9.3.1. and the the EC_50_ values were calculated.

**Table 1 antibodies-11-00013-t001:** Sequence features of the 34 unique scFvs shown in Figure 1. “Frequency” means the number of times a given sequence was found in the 53 positive clones. “VL” is the synthetic scaffold of the scFvs. “IGHV” is the germline gene determined by comparison with the human IGHV germline gene repertoire using IgBLAST (https://www.ncbi.nlm.nih.gov/igblast/; accessed on 5 January, 2022). “HCDR3 length” lists the number of amino acids per HCDR3 in that region (Kabat’s definition). Notice that, in C3 and C4, two different HCDR3 of 15 residues are reported, which are distinguished from each other with “15a” and “15b”. Highlighted in grey are the scFvs progressed to hIgG1 conversion.

Cluster	scFv	Frequency	VL Scaffold	IGHV Germline Gene	HCDR3 Length (aa)
1	G12	1	1-39	1-69	22
1	D11	1	1-39	1-69	22
1	D3	1	3-11	1-69	22
1	E1	1	ND	1-69	22
1	E10	1	3-20	1-69	22
1	G1	1	3-20	1-69	22
1	C10	1	4-01	1-69	22
2	A2	2	1-39	3-53	11
2	E2	1	1-39	3-53	11
2	A11	1	3-11	3-53	11
2	C2	1	1-39	3-53	11
2	B1	1	1-39	3-53	12
2	D9	1	1-39	3-53	12
2	A7	3	3-20	1-24	13
2	B8	1	3-20	1-24	13
3	B4	1	1-39	3-53	11
3	D7	1	1-39	1-46	15a
3	A3	8	3-20	1-46	15b
3	B7	3	3-20	1-46	15b
3	C4	3	3-20	1-46	15b
3	E5	1	3-20	1-46	15b
3	G9	1	3-20	1-46	15b
3	H6	2	4-01	1-46	15b
3	D4	5	4-01	1-46	15b
3	E9	1	3-20	3-23	17
3	B11	1	3-20	3-9	19
4	H2	1	1-39	1-46	15a
4	F1	1	1-39	1-46	15a
4	C6	1	3-20	1-46	15b
4	D10	1	3-20	1-46	15b
4	D2	1	1-39	1-46	15b
4	F4	1	3-20	1-46	15b
4	G2	1	3-20	1-46	15b
4	G3	1	3-20	1-46	15b

**Table 2 antibodies-11-00013-t002:** Developability profile of the Protein-A purified anti-SARS-CoV-2 antibodies. ^(a)^ The percent of monomer as determined by analytical SEC. ^(b)^ SDS-PAGE. Molecular weight as estimated in non-reducing (NR) and reducing (R) conditions. In the latter, the first number corresponds with the heavy chain and the second with the light chain. ^(c)^ Melting temperature (Tm) as determined by protein thermal shift assay. ^(d)^ Expression yield after four-day culture in adherents HEK 293T cells.

IgG	Monomer ^a^ (%)	SDS-PAGE ^b^		Tm ^c^ (°C)	Expression Yield ^d^ (mg/L)
		NR (kDa)	R (kDa)		
A2	100	140	49/25	71.3	19.92
A7	100	148	52/25	68.5 (81.8)	24.76
B1	100	158	48/25	71.9	15.82
G12	100	176	50/25	71.1	19.57

## Data Availability

The datasets generated for this study are available on request to the corresponding author.

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
