# Peer review of "Anti-SARS-CoV-2 Omicron Antibodies Isolated from a SARS-CoV-2 Delta Semi-Immune Phage Display Library"

_2073-4468, 2022, doi:10.3390/antib11010013_

Round 1

Reviewer 1 Report

This paper describes the generation of an immune, semi-synthetic scFv phage display library from a convalescent COVID19 patient. The patient had one dose of vaccine previously, and was infected with the Delta strain. The library was panned against the spike protein receptor binding protein (WT strain), and then clones screened against both the WT and Delta strains. This resulted in 34 different scFv sequences with properties that fell into four clusters. 7 were chosen for whole antibody production, the properties of which were consistent with the parent scFvs. One of the 7 antibodies was also able to bind the Omicron strain RBD protein. As this was also able to block spike binding to ACE2 protein, it has potential as a neutralising antibody therapy across multiple variants of SARS-CoV2.

The work describes the preliminary discovery step with all experiments being in vitro and using recombinant reagents. The authors should acknowledge in the discussion that the results need to confirmed on native sources of spike protein and tested in viral neutralisation assays.

The methodology used for library construction and biopanning is well described and standard procedures.  After biopanning, clones were screened after being expressed as scFv. scFv expression was confirmed by an assay requiring dual binding to both Protein L and Protein A. Protein L binding would be an intrinsic quality of the synthetic light chain repertoire, however it is not guaranteed that all heavy chains in the library would have Protein A binding ability. Would some clones be missed during screening due to their inability to bind Protein A? Why not detect with Anti-myc?

Protein A binding to the variable regions is reported to be restricted to Vh3 family, yet the project isolated Vh1 family after screening with Protein A. Was another screening method (eg Anti-myc) employed?

Figure 1:  Add titles to the four sections (Cluster C1, Cluster C2 etc)

Table 1: There is no need to duplicate the ELISA result in the table, or if it is left in then this needs to be described in the table headings or legend. I did not immediately realise that the last four columns were OD readings from the ELISAs in Figure 1.

Table 1: What is meant by the incomplete sentence 'The Different sequences of the same length' in the Table description - is this referring to the 'a' and 'b' additions to the CDR3 length?

Table 1: In the table description, change the final sentence to: 'In gray are highlighted the scFvs that were progressed to hIgG conversion.'

Line 285: Last word should be grey not green.

Line 96: Spelling error. Should be month not moth

Author Response

Comment: This paper describes the generation of an immune, semi-synthetic scFv phage display library from a convalescent COVID19 patient. The patient had one dose of vaccine previously, and was infected with the Delta strain. The library was panned against the spike protein receptor binding protein (WT strain), and then clones screened against both the WT and Delta strains. This resulted in 34 different scFv sequences with properties that fell into four clusters. 7 were chosen for whole antibody production, the properties of which were consistent with the parent scFvs. One of the 7 antibodies was also able to bind the Omicron strain RBD protein. As this was also able to block spike binding to ACE2 protein, it has potential as a neutralising antibody therapy across multiple variants of SARS-CoV2.The work describes the preliminary discovery step with all experiments being in vitro and using recombinant reagents. The authors should acknowledge in the discussion that the results need to confirmed on native sources of spike protein and tested in viral neutralisation assays.

Agree. Added in the last paragraph of the discussion: “Considering the fast evolution of SARS-CoV-2 into VOCs and the limited effectiveness of some vaccines to deal with the fast generation of the scape mutants as well as the lack of efficacy of some FDA and EMA EUA antibodies to treat SARS-CoV2 Delta and Omicron, if IgG-A7 demonstrates to neutralize the three SARS-CoV2 strains in vivo, it could certainly be a short-term solution to control the impact of new waves of SARS-CoV-2 infections.”

Comment: The methodology used for library construction and biopanning is well described and standard procedures.  After biopanning, clones were screened after being expressed as scFv. scFv expression was confirmed by an assay requiring dual binding to both Protein L and Protein A. Protein L binding would be an intrinsic quality of the synthetic light chain repertoire, however it is not guaranteed that all heavy chains in the library would have Protein A binding ability. Would some clones be missed during screening due to their inability to bind Protein A? Why not detect with Anti-myc?Protein A binding to the variable regions is reported to be restricted to Vh3 family, yet the project isolated Vh1 family after screening with Protein A. Was another screening method (eg Anti-myc) employed?

Agree. It was confusing in the early version of the manuscript. We used the Protein A/L assay to assess expression of the IgGs in HEK293 supernatants. For the scFvs we used Protein L/anti-myc. It is now corrected in Material & Methods (line 175-177) as follows: “The plates were washed with PBS and bound scFvs were detected with an anti-myc-HRP antibody (Abcam, Cat. No.: ab19312).”

Comment: Figure 1:  Add titles to the four sections (Cluster C1, Cluster C2 etc).

Done.

Comment: Table 1: There is no need to duplicate the ELISA result in the table, or if it is left in then this needs to be described in the table headings or legend. I did not immediately realise that the last four columns were OD readings from the ELISAs in Figure 1.

OD values removed from Table 1.

Comment: Table 1: What is meant by the incomplete sentence 'The Different sequences of the same length' in the Table description - is this referring to the 'a' and 'b' additions to the CDR3 length?

Corrected – see Table’s legend.

Comment: Table 1: In the table description, change the final sentence to: 'In gray are highlighted the scFvs that were progressed to hIgG conversion.'

Done.

Comment: Line 285: Last word should be grey not green.

Done.

Comment: Line 96: Spelling error. Should be month not moth

Done.

Reviewer 2 Report

In this manuscript by Ivette Mendoza-Salazar et al., the authors  have isolated and produced recombinant anti-SARS-Cov2 Omicron antibodies from a SARS-Cov-2 Delta semi-immune phage display library. The authors have identified triple cross-reactive antibody, named IgG-A7 that may neutralize SARS-coV2 wild-type, Delta and Omicron variants. 

In my opinion the research and methods design are appropiate and adeguate.

In particular, i emphasize the high scientific soundness of a semi-immune scFv phage display library with VH repertorie from a convalescent COVID-19 patient.

However the manuscript should be improved in the following parts:

1) In the material and methods section: 2.4 "Expression and purification of SARS-coV2 RBD recombinant proteins" the authors should be insert expression, purification, and characterization of the RBD-Omicron.

2) In the Results section the authors should be insert SEC analysis and SDS-PAGE and Protein Thermal shift assay for the candidate therapeutic antibody IgG-A7.  

3) In the Table 2. the authors should be insert the exact percent monomer was determined by SEC analysis, beacause in the line 335 the authors write:" monomeric content close to 100% after a single -step Protein A purification,"

4)In the conclusion section the authors should be insert considerations and assumptions about the epitope binding of the candidate therapeutic antibody IgG-7A.

Author Response

Reviewer 2:

Comment: In this manuscript by Ivette Mendoza-Salazar et al., the authors  have isolated and produced recombinant anti-SARS-Cov2 Omicron antibodies from a SARS-Cov-2 Delta semi-immune phage display library. The authors have identified triple cross-reactive antibody, named IgG-A7 that may neutralize SARS-coV2 wild-type, Delta and Omicron variants. 

In my opinion the research and methods design are appropiate and adeguate.

In particular, i emphasize the high scientific soundness of a semi-immune scFv phage display library with VH repertorie from a convalescent COVID-19 patient.

However the manuscript should be improved in the following parts:

1) In the material and methods section: 2.4 "Expression and purification of SARS-coV2 RBD recombinant proteins" the authors should be insert expression, purification, and characterization of the RBD-Omicron.

Agree. It was not clear in the early version of the manuscript. We did produce the RBDs Wuhan and Delta. We did not produce the RBD Omicron. In the revised version of the manuscript, we added the following text (line 148) to clarify this point: “The RBD Omicron was purchased at SinoBiological (Cat. No.: 40592-V08H121).”

Comment: 2) In the Results section the authors should be insert SEC analysis and SDS-PAGE and Protein Thermal shift assay for the candidate therapeutic antibody IgG-A7.  

We reported the IgG-A7 developability data in Table 2 – we can add the SEC, SDS-PAGE and Thermal-shift profiles in a supplementary material section but, in our opinion, it won’t add much to the results and discussion. We can share the information with the referee if necessary.  

Comment: 3) In the Table 2. the authors should be insert the exact percent monomer was determined by SEC analysis, beacause in the line 335 the authors write:" monomeric content close to 100% after a single -step Protein A purification,"

Agree. The monomeric content was 100% - it is now corrected in the text.

Comment: 4)In the conclusion section the authors should be insert considerations and assumptions about the epitope binding of the candidate therapeutic antibody IgG-7A.

We discuss the RBD:hACE2 blocking assay and competition with CB6 and P5E1. Both CB6 and ACE2 structures in complex with RBD Wuhan have been published and are available at the PBD. Therefore, we can have a Figure with this information and mention that the region recognized by A7 should overlap with the regions recognized by CB6 and ACE2. However, the details of the interaction would be speculative and hence, we prefer not include a detailed discussion of the interaction as it may not hold true once A7 epitope is elucidated experimentally.  

Reviewer 3 Report

Remarks to the Author:

In the manuscript ‘anti-sars-cov-2 omicron antibodies isolated from a sars-cov-2 delta semi-immune phage display library’, the authors isolated several antibodies recognizing sars2 RBD, and one of them showed cross-reacting with Omicron RBD, which is interesting. The submitted evidence is revealing application potentials of these antibodies in clinical.

The general comment here is please re-organize all figs to generate them in a more scientific way and facilitate the reading.

Major comments:

  1. line99, since tRNA is usually referring to transfer RNA, using tRNA for total RNA here is not appropriate
  2. for method 2.4. section, should also include the purification of Omicron RBD, since it was used in the last fig
  3. line141, based on the context before and after, should be RBD
  4. line195, in this method section, need to

(1) write down how the blocking data was calculated

(2) point out which RBD was used, WT? DT?

  1. for fig. 1

(1) please reorganize to facilitate the data interpretation, e.g., (1) label the four clusters clearly in the fig, (2) label the control clearly

this fig includes lots of key information, the presentation need to be improved and clear

(2) where are the two bold horizontal lines that indicate the cutoff value?

(3) for P5E1-A6 competition, why the negative control didn't show high binding? (the far right column)

(4) for ACE2 blocking assay, why the positive control colum is not visible?

  1. for table 1,

(1) the four columns at far right are not necessary, not making new points here, which have already been presented in fig 1

(2) why the control is showing '-'?

  1. for fig 2., please re-organize the fig, similar with fig 1., e.g., x/y axis title
  2. for table 2.

(1) not necessary to include VL/IGHV gene usage and HCDR3 length again here, which have already been showed in table 1

(2) un-recognized character under 'Tm'

  1. for fig 4., this data is not convincing. Considering the broad mutation all over the omicron, including the RBD, please either provide competition result against ACE2 on RBD-Omicron, Or, perform pseudotyped virus neutralization assay for at least the leading candidate, IgG-A7

Author Response

Reviewer 3

Remarks to the Author:

In the manuscript ‘anti-sars-cov-2 omicron antibodies isolated from a sars-cov-2 delta semi-immune phage display library’, the authors isolated several antibodies recognizing sars2 RBD, and one of them showed cross-reacting with Omicron RBD, which is interesting. The submitted evidence is revealing application potentials of these antibodies in clinical.

The general comment here is please re-organize all figs to generate them in a more scientific way and facilitate the reading.The Figures were improved (see comments to referee 1). Also, the order in which the antibodies are presented is the same in all the Figures so that the reading of the manuscript is more fluid.

Major comments:

  1. line99, since tRNA is usually referring to transfer RNA, using tRNA for total RNA here is not appropriate

Removed tRNA

for method 2.4. section, should also include the purification of Omicron RBD, since it was used in the last fig.

We did not produce the RBD Omicron. To clarify this point, we added in the revised version of the manuscript the following text in line 148: “The RBD Omicron was purchased at SinoBiological (Cat. No.: 40592-V08H121).”

  1. line141, based on the context before and after, should be RBD

Revised.

  1. line195, in this method section, need to
  • write down how the blocking data was calculated

Revised.

(2) point out which RBD was used, WT? DT?

It is specified in line 204 including the catalogue number.

  1. for fig. 1
  • please reorganize to facilitate the data interpretation, e.g., (1) label the four clusters clearly in the fig, (2) label the control clearly

Done.

this fig includes lots of key information, the presentation need to be improved and clear

  • where are the two bold horizontal lines that indicate the cutoff value?
  • for P5E1-A6 competition, why the negative control didn't show high binding? (the far right column)

Corrected.

(4) for ACE2 blocking assay, why the positive control colum is not visible?

  1. for table 1,

Corrected.

  • the four columns at far right are not necessary, not making new points here, which have already been presented in fig 1

Corrected – see also referee 1.

  • why the control is showing '-'?

Corrected

  1. for fig 2., please re-organize the fig, similar with fig 1., e.g., x/y axis title
  2. for table 2.

Revised.

  • not necessary to include VL/IGHV gene usage and HCDR3 length again here, which have already been showed in table 1

Removed

  • un-recognized character under 'Tm'

Corrected.

  1. for fig 4., this data is not convincing. Considering the broad mutation all over the omicron, including the RBD, please either provide competition result against ACE2 on RBD-Omicron, Or, perform pseudotyped virus neutralization assay for at least the leading candidate, IgG-A7

In progress a PRNT assay with A7 and Omicron - already completed for A7 with Wuhan and Delta, with good results. We are in the process of isolating SARS-CoV-2 Omicron. Once we got it, we will proceed with the PRNT for Omicron. We are planning on publish the results in a follow-up paper, together with other antibodies we are isolating from the library. In response to the referee’s concern, notice that we included in the revised version of the manuscript BIAcore data (lines 382 – 386) which, in our opinion, provides further support to the ELISA binding data.

Round 2

Reviewer 3 Report

Thank you for the response from the authors!